# Advances in the Radiological Evaluation of and Theranostics for Glioblastoma

**DOI:** 10.3390/cancers15164162

**Published:** 2023-08-18

**Authors:** Grayson W. Hooper, Shehbaz Ansari, Jason M. Johnson, Daniel T. Ginat

**Affiliations:** 1Landstuhl Regional Medical Center, 66849 Landstuhl, Germany; grayson.w.hooper.mil@health.mil; 2Rush University Medical Center, Department of Radiology and Nuclear Medicine, Chicago, IL 60612, USA; shehbaz_m_ansari@rush.edu; 3Department of Neuroradiology, The University of Texas MD Anderson Cancer Center, Houston, TX 77030, USA; jjohnson12@mdanderson.org; 4Department of Radiology, University of Chicago, Chicago, IL 60637, USA

**Keywords:** glioblastoma, GBM, glioma, high-grade glioma, CT, MRI, PET, theranostics, radiomics

## Abstract

**Simple Summary:**

This article reviews recent advances in diagnostic imaging and theranostics for glioblastoma. In particular, the goal of this article is to inform readers of advances related to CT, MRI, and PET for optimal assessment and treatment of patients with glioblastoma.

**Abstract:**

Imaging is essential for evaluating patients with glioblastoma. Traditionally a multimodality undertaking, CT, including CT cerebral blood profusion, PET/CT with traditional fluorine-18 fluorodeoxyglucose (^18^F-FDG), and MRI have been the mainstays for diagnosis and post-therapeutic assessment. However, recent advances in these modalities, in league with the emerging fields of radiomics and theranostics, may prove helpful in improving diagnostic accuracy and treating the disease.

## 1. Introduction

Glioblastoma is a high-grade CNS neoplasm previously known as glioblastoma multiforme (GBM). A recent change in nomenclature now reserves the name GBM for neoplasms that lack isocitrate dehydrogenase (IDH) and histone 3 mutations, among others [1]. These wildtype GBMs are the dominant subset of grade 4 diffuse astrocytomas and carry a worse prognosis than their IDH mutant counterparts (astrocytoma, IDH mutant, CNS WHO grade 4).

GBM is a notoriously lethal tumor with a mean survival time of 15 months, presenting diagnostic and therapeutic conundrums. The lethality of GBM is thought to be secondary to multiple intrinsic traits. The first is tumor heterogeneity [2]. Indeed, GBM demonstrates a marked degree of internal heterogeneity, which may render biopsies from one tumor segment utterly different from others. Therefore, information used to tailor a patient’s therapy may only reflect a part of the neoplasm. The second trait contributing to tumor lethality is its possession of tumor stem cells. These stem cells are thought to elucidate systemic immunosuppression, enhance tumor infiltrative capacity, and increase chemoresistance after initial therapy [3]. As modern medical research uncovers the specific genetics of GBM, the list of potential pharmacologic and immunologic treatments that may, one day, successfully manage or even cure this disease also grows. Consequently, diagnostic imaging must also advance with therapeutic advancements to remain relevant in this arena, as the tumor heterogeneity mentioned above makes repeat and/or multisite biopsies a risky endeavor.

Diagnostic imaging is critical in the workup and monitoring of patients with GBM. It first serves to delineate the extent of tumor involvement, formulate a differential diagnosis, and then help assess treatment response later. In years past, diagnostic imaging was largely qualitative, relying exclusively on the analysis of the radiologist and treatment team. However, recent CT, MRI, and artificial intelligence developments have added quantitative components to the imaging modalities. This quantitative information supplements subjective analysis and can provide insights regarding the deeper phenotype and genotype of the tumor, which has both treatment and prognostic implications. The field of theranostics, which involves the use of radioactive drugs with diagnostic and therapeutic potential (discussed below), has been making strides in league with molecular genetics and artificial intelligence. It is an emerging field that may provide diagnostic and therapeutic functionality for GBM. Recent advances in these topics are reviewed in the following sections.

## 2. Computed Tomography (CT)

Patients with GBM may undergo CT at initial presentation. CT typically reveals a low-attenuating, ill-defined tumor that can contain areas of high attenuation reflecting increased cellular density or tumoral hemorrhage if it has occurred. Postcontrast imaging may show ring enhancement in a closed configuration, though this is only sometimes the case. Standard CT is not very sensitive or specific for evaluating glioblastoma but can be augmented with cerebral perfusion mapping. Given that glioblastomas are highly vascular tumors with microvascular proliferation dependent upon tumor elaboration of growth factors like vascular endothelial growth factor (VEGF), it follows that blood flow to these lesions will be a distinguishing imaging characteristic. Specifically, tumoral cerebral blood flow (CBF), permeability, and cerebral blood volume (CBV) measurements tend to be elevated in glioblastoma (Figure 1), generally more so than in metastatic deposits and primary CNS lymphoma [4]. Furthermore, CT perfusion can also help differentiate WHO grade 3 gliomas from GBM, utilizing the permeability surface area product and cerebral blood volume [5].

CT scans can be computationally analyzed beyond the threshold of human perception to extract information about a tumor’s attenuation, shape, heterogeneity, and other semantic and agnostic features, known as CT radiomics. For example, a 2013 study showed that tumor heterogeneity on CT could help distinguish between high- and low-grade gliomas with coarser features more indicative of higher grades [6]. A second study showed that a seven-feature textural analysis of the peritumoral zone of an intra-axial mass could help differentiate between a high-grade glioma and a brain metastasis with statistical significance [7]. This retrospective study tested the hypothesis that the peritumoral zones of gliomas would have “busier” textural analyses versus metastatic deposits. Moreover, as suggested by the infiltrative growth of gliomas, the peritumoral zones of these lesions demonstrated higher pixel intensity, lower homogeneity, and higher randomness. Thus, valuable prognostic and phenotypic information can be obtained by analyzing agnostic textural features on CT.

CT has a distinct advantage over MRI in radiomics, as Hounsfield Units (HU) correlate well with tissue attenuation [8]. CT acquisition is also much faster than MRI, which helps mitigate motion artifacts, a known problem that can diminish the reproducibility of essential features. However, there are limitations to radiomics regardless of modality. These include interobserver variability during imaging segmentation, as some regions of interest must be manually drawn, and equipment variability. Another limitation is that current radiomics models are based chiefly on retrospective studies, which must be standardized to optimize their value for future diagnostic use.

## 3. Magnetic Resonance Imaging (MRI)

When compared to CT, MRI generally provides better anatomic detail and tumor infiltration characteristics. Despite these advantages, standard postcontrast T1-weighted MRI cannot readily distinguish the histologic grade of gliomas, nor can it consistently differentiate from several potential mimics of GBM, such as primary CNS lymphoma, other histologic primary brain neoplasms, tumefactive multiple sclerosis, stroke, and even solitary toxoplasmosis [9,10]. Nevertheless, diffusion-weighted imaging (DWI), MRI spectroscopy, and MRI radiomics can improve diagnostic accuracy.

DWI of a suspected glioblastoma with ADC mapping can provide important information about cellular density. Analysis of the resulting histogram may hold clues in differentiating abscesses from high-grade gliomas and grade 3 gliomas from GBM. ADC histogram analysis is a method of quantifying the heterogeneity of diffusivity within a lesion and has shown promise both in the brain and abdomen [11,12]. A 2018 study investigating ADC histograms showed that profiling GBMs and brain abscesses, specifically ADCp10, could differentiate the two with statistical significance [13]. Another study in 2021 showed that analysis of first- and second-order values from ADC histograms could be used to differentiate grade 3 from grade 4 gliomas and that a significant inverse correlation exists between these values and Ki-67 expression [14].

Diffusion tensor imaging (DTI) is a technique that is also useful in the setting of GBM for assessing recurrence, classifying neoplasms, and surgical planning. DTI measures directional water diffusivity and is thus an indirect method of mapping white matter tracts. Furthermore, as studies have shown that gliomas migrate along these white matter tracts, it follows that changes in white matter microstructure may be assessed on DTI [15]. Indeed, Jin and colleagues demonstrated that serial DTI exams showed fractional anisotropy (FA) and neurite density index declines within white matter tracts in patients with GBM recurrence up to two months prior to findings on standard MRI [16]. DTI showed mixed results in delineating glioblastoma from primary CNS lymphoma, as FA was significantly lower in primary CNS lymphoma than GBM in one study and significantly higher in another [17,18]. A recent study by Razek and colleagues demonstrated a higher FA in primary CNS lymphoma versus glioblastoma. The combined metrics of tumor blood flood, mean diffusivity, and FA had 95.5% accuracy for differentiating the disease processes [19].

Similarly, other studies showed no significant difference in intratumoral FA between high-grade gliomas and solitary metastatic deposits. However, Holly and colleagues did show a significant increase in FA in the peritumoral zone of HGGs versus metastases [20,21]. In a 2020 meta-analysis by Zhang and Liu that included 19 studies, DWI and DTI showed “moderate diagnostic value” in differentiating glioblastomas from solitary brain metastases [22]. DTI tractography may also help map white matter tracts prior to surgery, and its use has resulted in changes to surgical approaches and improvements in patient recovery [23].

Dynamic susceptibility contrast perfusion-weighted imaging (DSC-PWI) is a magnetic resonance imaging (MRI) technique that can be used to assess the microvascular environment of brain tumors (Figure 2). DSC-PWI can be used to measure parameters such as CBV, CBF, and mean transit time (MTT). These parameters can be used to characterize tumors, assess response to treatment, and predict prognosis. DSC-PWI has also been shown to be useful in the diagnosis of brain tumors, particularly in cases where conventional MRI findings are inconclusive. In addition to diagnosis, DSC-PWI can also be used to assess response to treatment. For example, DSC-PWI can be used to measure the change in CBV after radiation therapy [24]. This information can be used to monitor the effectiveness of treatment and to adjust treatment plans as needed. Finally, DSC-PWI can also be used to predict prognosis. For example, studies have shown that patients with high CBV are more likely to have a poor prognosis [25]. This information can be used to counsel patients and to make decisions about treatment. Overall, DSC-PWI is a valuable tool for the diagnosis, assessment of treatment response, and prediction of the prognosis of brain tumors [26].

MRI spectroscopy (MRS) measures the resonance frequencies of compounds, including biological metabolites. For clinical purposes, 1H (proton) and phosphorus 31 (31P) resonances are used to measure myoinositol, choline, creatine, n-acetyl aspartate (NAA), and lipids/lactate, as well as various phosphorus metabolites. As such, this technique can glean additional information about an active region of interest when compared against a standard control—often the contralateral, uninvolved hemisphere (Figure 3). Typically, MRS supplements diagnostic information and can be helpful in disease processes and tumor types, though significant overlaps in metabolite profiles are possible (Table 1). It is also helpful in delineating radiation necrosis from pseudoprogression and actual disease progression, as an elevated choline/NAA ratio has a high sensitivity and specificity for actual disease progression [27,28,29]. However, more recently, MRS has been used to search for genotype-specific metabolites, which help identify specific mutations. One such metabolite is 2-hydroxygluarate (2-HG), which accumulates after mutation in IDH1 and/or IDH2 [30]. This metabolite is highly specific for such mutations and rarely encountered aside from hydroxyglutaric aciduria [31]. Indeed, as higher field strengths are vetted for safe clinical use, the potential for more diverse MRS metabolite detection improves. Higher magnetic field strengths improve the spectral separation of specific metabolites and enhance signal-to-noise ratios [32,33].

Chemical exchange saturation transfer (CEST) is a relatively new, noninvasive MRI technique with glioblastoma applications. Similar to 1H MRS, CEST measures the resonance frequencies of protons; however, CEST broadly quantifies the signal generated by specific groups (amide, amines, hydroxyl groups, etc.) after magnetization transfer from their protons to those within the surrounding water solute body [34]. As a result, the concentrations of proteins and other biologically relevant solvents can be assessed, which can be used to predict various features of a CNS neoplasm.

In a prospective study quantifying the nuclear Overhauser effect and amide-proton-transfer-weighted (APTw) values in a small cohort of brain tumors, Paech et al. demonstrated significant differences between low-grade and high-grade gliomas as well as IDH wildtype versus mutant [35]. Specifically, a more significant amide signal was seen in higher-grade gliomas, consistent with the logical assertion that more aggressive tumors expressed greater protein concentrations. Further, the *lower* signal was seen in IDH mutant tumors, which is also consistent with this gene mutation that downregulates protein expression downstream [36]. Another promising aspect of CEST in glioma is the use of d-glucose as a contrast agent. Measuring the saturation exchanges in the hydroxyl groups on glucose, studies have shown activity corresponding to gadolinium contrast enhancement in brain tumors. This information suggests that the exogenous glucose can localize to regions of the breakdown of the blood–brain barrier and be detected on 3T magnets [37,38].

APTw imaging at 3.5 ppm will be hyperintense in regions of increased protein and cellular turnover and helps differentiate various neoplasms from ordinary appearing white matter, primary brain tumors from metastases, and monitor treatment responses [39]. Given that GBM is a heterogeneous tumor, protein concentrations will vary throughout the bulk of the mass, which may be secondary to regional protein expression differences or hemorrhage. However, other disease processes can also create a protein-rich milieu and cause hemorrhage. It is here, too, that radiomics may prove helpful, as both standard and advanced MRI imaging sequences may also be subject to radiomic analysis. Concerning APTw CEST radiomics, proof-of-concept studies have yielded promising results when differentiating GBM from lower-grade gliomas and brain metastases, as well as the specific phenotype of a brainstem glioma [40,41].

The genetic variations of glioblastomas have been found to correlate with prognosis and treatment response and represent targets for future therapies, with MGMT promoter methylation status being vital [42]. The current gold standard of diagnosis is invasive, requiring tissue acquisition for analysis, but radiomics has the potential to provide rapid insights regarding tumor histology and genomics. Based on deep learning computational techniques, Kobayashi and colleagues could predict glioma grades with 90% accuracy [43]. Several studies have shown promise in predicting tumor biomarkers relevant for prognosis and treatment, such as MGMT, IDH, and TERT [44,45,46,47]. Moreover, adding a radiomics model to the clinical and genetic profiles of GBM patients has improved the prediction of progression-free survival and overall survival [48].

Though significantly more complex than the density values of CT, MRI images contain a wealth of information beyond CT. However, MRI radiomics has its unique challenges. From greater susceptibility to motion artifacts, field inhomogeneity, and a plethora of technical factors, imaging features on this modality are prone to variability and are challenging to reproduce. Nevertheless, machine learning algorithms have been applied to many prior studies on GBM patients to seek useful features and build models. These models have suggested good correlations for prognosis and have elucidated the tumor phenotype and even genotype. Unfortunately, radiomics models thus far have been mainly generated from retrospective data and have limited reproducibility.

Additionally, the field may be prone to errors intrinsic to high-level statistical analysis—specifically data leakage. A 2022 study by Gidwani and colleagues suggested that many contemporary radiomics studies report inflated values for their machine learning model accuracies. In this study, the team showed that model accuracy was spuriously elevated by a factor of 1.4 after randomly selecting 50 studies conducted between 2017 and 2021 for analysis. They deduced that errors in data partitioning and selecting unproductive radiologic features contributed to this inflation. When such was corrected, the resulting simulated models showed areas under the receiver operating characteristic curve of 0.5—random chance [49]. As a result, it may be some time before radiomics models gain clinical acceptance and utility, as large, multicenter, prospective trials will be required for vetting.

## 4. Positron Emission Tomography (PET)

PET utilizes either ^18^F-FDG, a glucose analog, or labeled amino acids with preferential tumor uptake for glioblastoma imaging. A selection of radiotracers with long and complex names has demonstrated potential in imaging glioblastoma (Table 2). These radiotracers are analogs of components the tumor needs for energy production, protein synthesis, DNA synthesis, or cellular communication. Other radiotracers are currently experimental and seek to bind to sigma-1 or -2 receptors or specific enzymes.

^18^F-FDG is valuable in the diagnosis and prognosis of glioblastoma; however, interpretation can be imprecise if the tumor is adjacent to gray matter, which is also highly metabolically active [50,51]. Delayed imaging acquisition, however, has been shown to provide superior conspicuity of glioblastoma compared to conventional image acquisition times and can help to differentiate normal cortical hypermetabolism from high-grade lesions (Figure 4) [52]. Regardless, ^18^F-FDG can still differentiate HGGs from other tumor types. When accounting for all other variables, the tumor-to-normal brain tissue ratio is the most significant predictor of progression-free survival and overall survival. ^11^C-methyl-methionine (^11^C-MET) has been shown to have superior sensitivity to ^18^F-FDG and MRI in delineating tumor extent and demonstrates utility in evaluating treatment response. Aside from the nontrivial challenge of crossing the blood–brain barrier, multiple agents have demonstrated nonspecific binding that engenders a low signal-to-noise ratio. Others, like those targeting sigma receptors, can bind opiate receptors due to structural similarities [53].

PET may also help differentiate actual progression from pseudoprogression [54]. Concerning standard PET, ^18^F-FDG and ^11^C-MET detect recurrent tumors similarly [55]. However, PET-CT performed with ^11^C-MET and subjected to radiomics random forest classification demonstrated a 90.1% sensitivity and 93.9% specificity in differentiating glioma recurrence from radiation necrosis [56]. [18F]-L-dihydroxyphenylalanine (18F-FDOPA) is also an amino acid analog that is preferentially taken up by gliomas but has a higher sensitivity in detecting recurrent disease than ^18^F-FDG [57,58,59]. It has also shown higher SUVmax, SUVmean, tumor-to-normal brain, and tumor-to-striatum values with a diagnostic accuracy of 82% in delineating progression- vs. treatment-based changes [60]. 18F-fluciclovine (18F-FACBC) is an FDA-approved radiotracer for prostate cancer imaging with multiple publications demonstrating its capability in imaging gliomas. FACBC PET has high diagnostic accuracy in defining tumor extent, volumes, and satellite lesions better than MR; compared to methionine, FACBC has similar accuracy but better tumor-to-background contrast; FACBC uptake may help to discriminate between low-grade and high-grade glioma. Radiolabeled fluciclovine (18F-FACBC) imaging seems to be useful in analyzing glioma/glioblastoma [61]. Amino acid PET can detect changes in how tumor cells use energy before those changes can be seen on MRI. This allows for the determination of whether therapy is working earlier and more accurately than MRI techniques alone. Amino acid PET uptake is also specific to tumor cells, so it can help to distinguish between tumor tissue and other types of tissue, such as inflammation or edema. This information can help to improve the accuracy of diagnosis and treatment planning. These therapy-induced metabolic changes detected via amino acid PET facilitate early treatment response assessments. Integrating amino acid PET in the management of CNS malignancies to complement MRI will significantly improve early therapy response assessment, treatment planning, and clinical trial design [62].

GBMs have a hypoxic microenvironment due to the rapid expansion of the solid tumor and the inefficient, tangled, and abnormal blood vessels they recruit. As a result, these tumors preferentially utilize pyruvate in aerobic glycolysis for their energy supply (Warburg Effect). Hypoxia is thought to drive GBM invasiveness via the upregulation of ephrin type-A receptor 2 (EphA2), among others, which codes for the promotility receptor tyrosine kinase on GBM cells. It also contributes to the expression of hypoxia-inducible factor (HIF), a key regulator of angiogenesis. Worse, mutations common in some GBMs can worsen HIF expression, specifically an epidermal growth factor receptor (EGFR) mutation and a phosphatase and tensin homolog (PTEN) mutation [63]. Hypoxia is also thought to drive chemoresistance in GBMs, with an excellent overview of the topic provided by Olivier and colleagues [64]. Finally, hypoxia is responsible for radiotherapy resistance, as it inhibits the creation of reactive oxygen species. With all of this in mind, localizing regions of hypoxia within GBMs would be beneficial in planning radiation therapy and future tailored therapies. ^18^F-FMISO-PET can localize regions of hypoxia and perfusion and has been used in conjunction with evofosfamide, an agent that preferentially activates in hypoxic environments, in tumor therapy [65]. Moreover, although a similar study was published in 2016 targeting rodent glioma models, the more recent literature is sparse [66].

## 5. Theranostics

Some of the most promising targets for theranostics are tumor-associated myeloid cells (TAMCs), which are both immune-inhibitory and, with glioma-associated microglia/macrophages (GAMMs), comprise up to 40% of a glioma’s (GBM’s) cellular mass [67]. GAMMs, consisting of infiltrating blood-derived macrophages and native CNS microglia, also contribute to immunosuppression via several mechanisms. These cells themselves are not malignant and form what can be considered a “logistics and support” engine for the malignancy. As such, they maintain the expression of various cell-surface receptors, which presents opportunities for future research. One especially promising target is CD11b. A 2021 study published by Foster and colleagues showed that a bifunctional chelator anti-CD11b antibody conjugated with Zr-89 for PET imaging and Lu-177 for targeted radiotherapy could localize to a glioma as well as target circulating CD11b+ cells in a mouse model. The result was a reduction in TAMCs and improved animal survival times after immune checkpoint inhibitor immunotherapy [68].

Overcoming the challenge of the blood–brain barrier (BBB) is essential for treating CNS neoplasms. This physical barrier, augmented via molecular pumps like p-glycoproteins, limits most conventional therapies’ effectiveness. However, one strategy to overcome this issue employs intravenous microbubbles that can be conjugated with various radiotracers and therapeutic agents or utilized purely for transient disruption of the BBB. Although the precise mechanism is unknown, focused ultrasound can cause bubble cavitation that disrupts tight junctions of the BBB and downregulates the expression of p-glycoprotein.

Preclinical trials have been promising, showing increased brain concentrations of standard chemotherapeutic agents and the monoclonal antibody bevacizumab in animal models [69]. Further, microbubbles have been used to reversibly disrupt the BBB to measure PET radiotracer accumulation with promising results [70]. Indeed, GBM is an infiltrating malignancy that can be underestimated with standard imaging modalities due to the relative preservation of the BBB at the periphery of the neoplasm relative to its core. Thus, transient disruption of the BBB with or without direct delivery of a tumor-specific radiotracer could provide invaluable preoperative information and/or response-to-therapy information.

Another promising means of both penetrating the BBB and delivering therapeutic agents is gold nanoparticles. These dynamic particles are bioinert and effective in penetrating the BBB if sized below 30 nm and can be PEGylated to increase longevity within the body [71]. Gold nanoparticles may also be conjugated with various moieties, including chemotherapeutic agents, immunotherapies, and gadolinium, to improve drug delivery and imaging. Furthermore, gold nanoparticles alone have been shown to localize within human glioblastoma cell lines and promote the radiosensitization of sensitive and resistant cells [72]. Chelated gold and gadolinium nanoparticles were also recently shown to cause radiosensitization of in vitro human glioblastoma cells and diminish their ability to invade surrounding brain parenchyma [73].

## 6. Future Directions

It is difficult to prognosticate the future of GBM treatment due to the possibility of the development of disruptive technology. With efforts in therapeutic development varying widely from novel chemotherapy agents and the revamping of old agents to developing particular monoclonal antibodies, it could emerge that GBM has an “Achilles’ Heel” that could allow curative therapy. However, such is unlikely, given the heterogeneous tissue composition of the disease. More logical would be a multipronged regimen targeting the tumor directly, the tumor microenvironment, stem cells, and the host’s immune system. This is more in keeping with what we know about the disease. A positive result would likely entail a significant extension of the patient’s quantity (and hopefully quality) of life. Regardless, diagnostic imaging will need to advance in kind to remain relevant. MRI and PET appear to be the modalities with the most significant potential to definitively diagnose the disease, possibly at the genetic level—the holy grail being a so-called radiologic biopsy or “radiopsy.” Indeed, with advancements in magnetic field strength granting greater contrast resolution to images and spectroscopy peaks, and the continuous development of PET radiotracers that can be both diagnostic and therapeutic, the future looks bright.

## 7. Conclusions

Glioblastoma remains a terrible disease with a dismal prognosis. This is because the tumor is markedly heterogeneous, presenting hurdles to diagnostic accuracy, and the tumor possesses stem cells, which impart significant resilience to all current forms of therapy. However, modern medicine is quickly advancing. The need for accurate, discriminating, and tailored diagnostic capabilities has increased with the discovery of new genetic mutations that impart phenotypic differences to the tumor and advancements in immunotherapy and radiotherapy. Therefore, it is of the utmost importance that diagnostic radiology evaluates, tests, and embraces many emerging technologies and techniques in the relevant modalities.

## Figures and Tables

**Figure 1 cancers-15-04162-f001:**
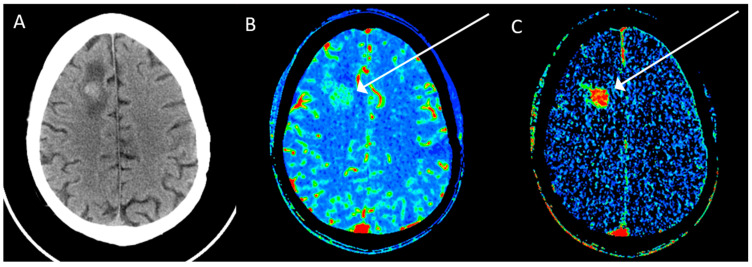
Axial CT image (**A**) shows a right frontal tumor that represents a glioblastoma with corresponding elevated blood volume (**B**) and elevated permeability (**C**) on CT perfusion (arrows).

**Figure 2 cancers-15-04162-f002:**
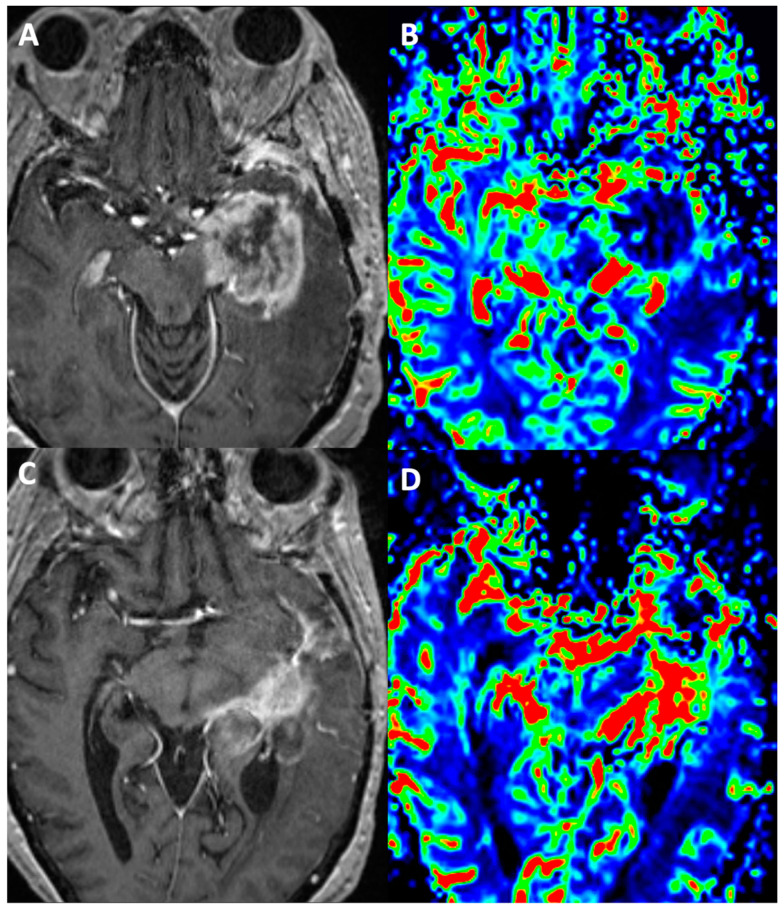
Pseudoprogression (**A**,**B**): Four months post gross total resection and completion of concomitant chemoradiation with temozolomide, this patient with left temporal lobe glioblastoma, MGMT unmethylated, shows nodular enhancement along the resection margins (**A**). The enhancing component shows no raised perfusion on rCBV maps obtained using DSC-PWI (**B**). Repeat surgical resection due to mass effects revealed post-treatment changes in the form of gliosis, radiation necrosis, and focal reactive spindle cell proliferation. True progression (**C**,**D**): Eight months post gross total resection and concomitant chemoradiation with temozolomide, this patient with glioblastoma, MGMT methylated, shows nodular enhancement in the left posterior temporal resection bed (**C**) with corresponding raised rCBV values on perfusion imaging (**D**). The imaging findings were suggestive of tumor recurrence, which was later confirmed on histopathology.

**Figure 3 cancers-15-04162-f003:**
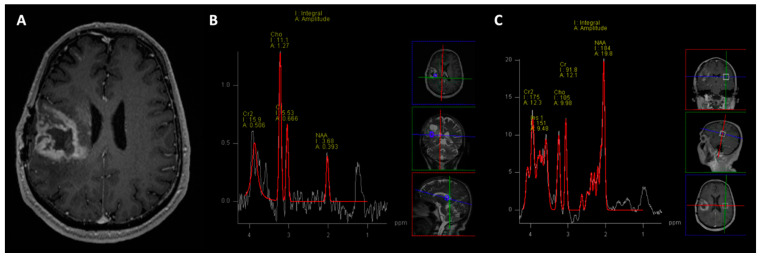
True progression: Two years following gross total resection and chemoradiation with temozolomide, this patient with right temporal glioblastoma, MGMT unmethylated, shows peripheral nodular enhancement along the resection margins (**A**). Multivoxel 1H MR spectroscopy performed at long TE (135 ms) shown in the NAA (n-acetylcysteine) values, and elevated choline peak with increased choline/creatinine and choline/NAA ratios in the enhancing regions (**B**), suggesting tumor recurrence, which was later confirmed on histopathology. C–MR spectroscopy on the corresponding contralateral brain parenchyma serves as control (**C**).

**Figure 4 cancers-15-04162-f004:**
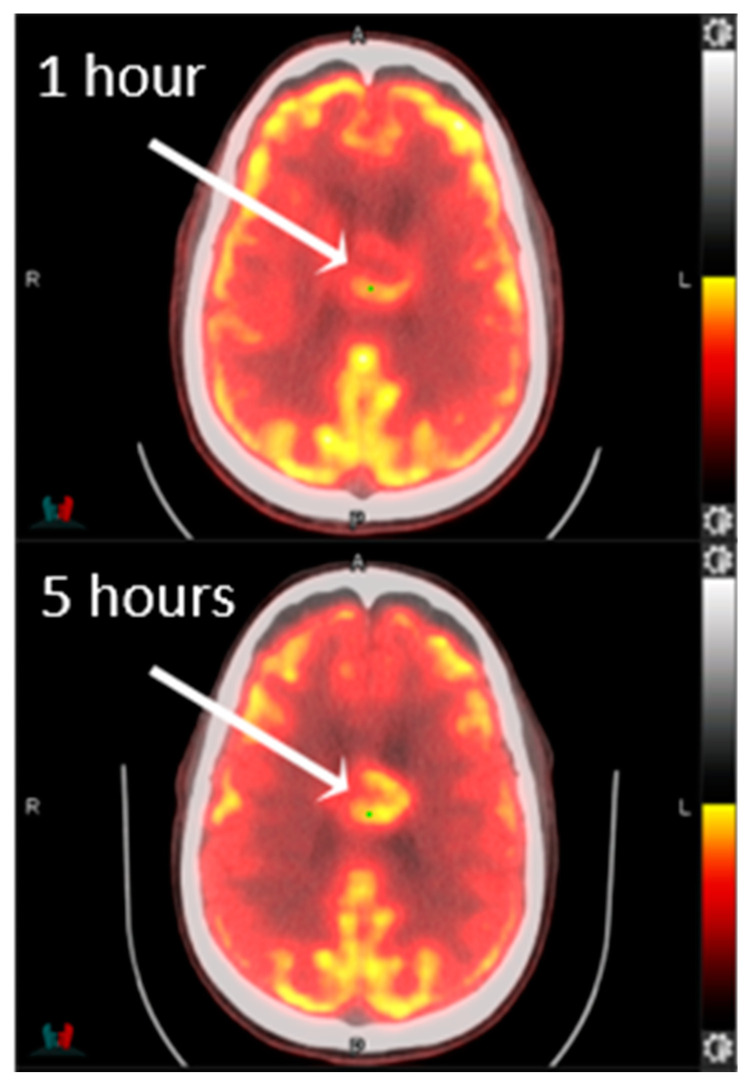
FDG-PET/CT images show greater conspicuity of the corpus callosum glioblastoma at 5 h than at 1 h.

**Table 1 cancers-15-04162-t001:** MRI spectroscopy of common disease processes and glioblastoma.

Disease	Myoinositol	Creatine	Choline	NAA	Lipid-Lactate	Chol/Cr	Chol/NAA
GBM	↓	↓	↑	↓	↑ ↑	>2.5	>2.2
CVA		↓		↓	↑ (Lactate)		
TMS	-	↓	↑	↓	↑ (Lactate)	↑	↑
Oligodendroglioma	↑	↓	↑	↓	↑↑	↑	↑
Metastasis	-	-	↑	↓	↑	↑ > 1.24	↑ > 1.11

CVA—stroke; TMS—tumefactive multiple sclerosis. There is significant overlap in the metabolic profiles of these disease processes. (↑—elevated; ↓—decreased).

**Table 2 cancers-15-04162-t002:** Potential radiotracers used for imaging glioblastoma.

Radiotracer	Function
^18^F-FDG	Glucose analog
[^11^C]Methionine ([^11^C]MET)	Amino acid preferentially utilized by gliomas vs. normal brain tissue; very short half-life.
[^18^F]L-fluoro-dihydroxyphenylalanine ([^18^F]FDOPA)	Amino acid similar to CMET but longer half-life.
[^18^F]Fluoromisoinodazole ([^18^F]FMISO)	Hypoxia-sensing agent, poor specificity. Low BBB penetration.
[^18^F]-fluorocyclobutane-1-carboxylic acid (18F-FACBC, Fluciclovine)	FDA-approved amino acid trace for prostate with performance similar to [^11^C]MET.
[^18^F]fluoroethyl-tyrosine ([^18^F]FET)	Actively transported; highly valuable when paired with MRI.

## Data Availability

Not applicable.

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
