# Peer review of "Advances in the Radiological Evaluation of and Theranostics for Glioblastoma"

_cancers, 2023, doi:10.3390/cancers15164162_

Round 1

Reviewer 1 Report

The term Theranostics should be defined.

Colloquialisms such as “muddled”, “ripe for research” should be replaced by academic English.

The WHO classification should not only be mentioned but followed:

“These wild-type GBMs are the dominant subset of grade 4 astrocytomas 26  and carry a worse prognosis than the mutant-type GBMs. Regardless, GBM is still used 27  interchangeably to discuss both tumors.”
(Type of the Paper (Article, p.1)

The latter is, of course, incorrect and this should be pointed out!

“… perfusion can also help differentiate WHO 67 grade 3 gliomas (anaplastic astrocytoma) from GBM utilizing the permeability surface 68 area product and cerebral blood volume [5].“

'Grade 3 gliomas' is NOT synonymous with anaplastic astrocytoma.

Please check this sentence (it does not seem to make sense):

“The patient underwent repeat 157  resection only to confirm tumor recurrence on histopathology.“
(Type of the Paper (Article, p.4)

There is a problem with the formatting of the table.

“Delayed imaging time point however have been shown” – points?

“tumor-associated myeloid 314  cells (TAMCs), which are both immune inhibitory and, with their derivative tumor-asso- 315  ciated macrophages (TAMs),
(Type of the Paper (Article, p.8)”

Microglia should be mentioned and discussed briefly.

“However, such is unlikely, given the heterogeneity of the disease
(Type of the Paper (Article, p.9)

The tissue composition is heterogenous rather than the disease.

The statement “However, we are not ruling 361 out a cure.”  does not really fit with the conclusion "the future looks 368  bright."
(Type of the Paper (Article, p.9)

It could become a chronic and manageable process, much like diabetes.
(Type of the Paper (Article, p.9)

This is an extremely odd comparison which would be better removed.

and the tu- 372  mor possesses stem cells
(Type of the Paper (Article, p.10)

The latter are not limited to glioblastoma.

see above

Author Response

  1. The term Theranostics should be defined: We have added the phrase “The field of theranostics, which involves use of radioactive drugs with diagnostic and therapeutic potential (discussed below), has been making strides in league with molecular genetics and artificial intelligence” as an attempt to explain the meaning of the word.
  2. Colloquialisms such as “muddled”, “ripe for research” should be replaced by academic English: We have replaced these words in the text.
  3. The WHO classification should not only be mentioned but followed: “These wild-type GBMs are the dominant subset of grade 4 astrocytomas 26 and carry a worse prognosis than the mutant-type GBMs. Regardless, GBM is still used 27 interchangeably to discuss both tumors.” The latter is, of course, incorrect and this should be pointed out!: we have reframed the introduction with use of terms congruent with the recent 2021 WHO classification of tumors of the CNS.
  4. “… perfusion can also help differentiate WHO 67 grade 3 gliomas (anaplastic astrocytoma) from GBM utilizing the permeability surface 68 area product and cerebral blood volume [5].“ 'Grade 3 gliomas' is NOT synonymous with anaplastic astrocytoma: The cited study had used 2007 WHO classification which was based on histological characteristics and grade 3 astrocytoma was synonymous with anaplastic astrocytoma in that classification. The phrase “anaplastic astrocytoma” has been removed in the updated manuscript to avoid confusion.
  5. Please check this sentence (it does not seem to make sense): “The patient underwent repeat 157 resection only to confirm tumor recurrence on histopathology: We apologize for the confusion. The statement has been reframed as “The imaging findings were concerning for tumor recurrence, which was later confirmed on histopathology” for better comprehension.
  6. There is a problem with the formatting of the table; Please check Table 1 in the main text, there are two tables in the main text, but only one table title: titles were added to both the table, table 1 was cited in the text and table 1 was moved closer to the cited text as per author instructions on the journal website.
  7. “Delayed imaging time point however have been shown” – points?: the word “points” have been replaced by “acquisition/acquisition time” in the text.
  8. “tumor-associated myeloid cells (TAMCs), which are both immune inhibitory and, with their derivative tumor-associated macrophages (TAMs), Microglia should be mentioned and discussed briefly: information on glioma-associated microglia/macrophages (GAMMs) has been added in the “theranostic” section of the manuscript.
  9. However, such is unlikely, given the heterogeneity of the disease. The tissue composition is heterogenous rather than the disease: The statement has been reframed as “However, such is unlikely, given the heterogeneous tissue composition of the disease.”
  10. The statement “However, we are not ruling 361 out a cure.”  does not really fit with the conclusion "the future looks 368 bright.” It could become a chronic and manageable process, much like diabetes. This is an extremely odd comparison which would be better removed: These statements have been removed from the text, as suggested.
  11. and the tumor possesses stem cells. The latter are not limited to glioblastoma: Although, tumor stem cells are not unique for glioblastoma, but are major component of the tumoral tissue. Hence, the statement has been rephrased as “This is because the tumor is markedly heterogeneous, presenting hurdles to diagnostic accuracy, and the tumor possesses stem cells, which impart significant resilience to all current forms of therapy.”

Reviewer 2 Report

In this manuscript,

Hooper et al. write an extensive review, on the topic of the Radiological Evaluation and Theranostics for Glioblastoma. This is a hot topic in the field as the development of more precise diagnostic ways to identify and screen GBM is potentially a new powerful tool to contrast tumor cancer progression. Recent advances in standard imaging modalities such as  CT, including CT cerebral blood perfusion, PET/CT may serve as a striking improvement  in diagnostic assessment and early treatment of the disease. 

The review covers a large and up-to-date amount of work and presents some interesting outlook for new  diagnostic and therapeutic applications. 

The manuscript is well written and has a clear flow in the description of the different techniques. 

Consequently I recommend this review for publication in this journal.  

Author Response

Thank you.